# GEOMETRY-BASED END-TO-END SEGMENTATION OF CORONARY ARTERY IN COMPUTED TOMOGRAPHY ANGIOGRAPHY

**Xiaoyu Yang** [1,2,*] **Lijian Xu** [2,3] [✉] **, Simon Yu** [4] **, Qing Xia** [5] **, Hongsheng Li** [6] **& Shaoting Zhang** [2]

[1] College of Electronics and Information Engineering, Tongji University
[2] Shanghai Artificial Intelligence Laboratory
[3] Centre for Perceptual and Interactive Intelligence, the Chinese University of Hong Kong
[4] Department of Imaging and Interventional Radiology, the Chinese University of Hong Kong
[5] SenseTime Research
[6] Department of Electronic Engineering, the Chinese University of Hong Kong

## ABSTRACT

Coronary artery segmentation has great significance in providing morphological information and treatment guidance in clinics. However, the complex structures with tiny and narrow branches of the coronary artery bring it a great challenge. Limited by the low resolution and poor contrast of medical images, voxel-based segmentation methods could potentially lead to fragmentations of segmented vessels and surface voids are commonly found in the reconstructed mesh. Therefore, we propose a geometry-based end-to-end segmentation method for the coronary artery in computed tomography angiography. A U-shaped network is applied to extract image features, which are projected to mesh space, driving the geometry-based network to deform the mesh. Integrating the ability of geometric deformation, the proposed network could output mesh results of the coronary artery directly. Besides, the centerline-based approach is utilized to produce the ground truth of the mesh instead of the traditional marching cube method. Extensive experiments on our collected dataset CCA-520 demonstrate the feasibility and robustness of our method. Quantitatively, our model achieves Dice of 0.779 and HD of 0.299, exceeding other methods in our dataset. Especially, our geometry-based model generates an accurate, intact and smooth mesh of the coronary artery, devoid of any fragmentations of segmented vessels.

## 1 INTRODUCTION

Segmentation of the coronary artery tree in coronary computed tomography angiography (CCTA) is of great clinical value, such as presenting the morphology of the coronary artery, exhibiting the lesion and guiding clinical treatment. However, the automatic segmentation of the coronary artery indicates a severe challenge. First of all, the coronary artery has a distinctive tree structure with tiny and narrow branches that vary dramatically. Some branches are too thin to be segmented accurately, especially interfered with by other blood vessels. Second, the sparsity and anisotropy of CCTA images result in most segmentation methods being voxel-based. The reconstructed mesh from the voxel-based segmentation mask is rough with an obvious lattice shape. Additionally, limitations of CCTA images, such as low resolution and poor contrast, make it more challenging to segment the coronary artery.

Recently, deep learning has shown its viability of coronary artery segmentation with high accuracy. Most current methods perform voxel-based segmentation and achieve improvements based on the Unet, such as 3D-FFR-Unet Song et al. (2022), TETRIS Lee et al. (2019), FFNet Zhu et al. (2022), PDS Zhang et al. (2022) and TreeConvGRU Kong et al. (2020). Instead of traditional voxel-based segmentation, mesh-deformation-based methods have been increasingly drawing the attention of the community. Voxel2Mesh Wickramasinghe et al. (2020) extends pixel2mesh Wang et al. (2018) to

---

*This work is done during an internship at Shanghai Artificial Intelligence Laboratory.

3D images for segmentation tasks of the liver, synaptic junction, and hippocampus. GMB Wang et al. (2021) exploits point net to refine voxel-based segmentation results of the coronary artery by removing irrelevant vessels, where point cloud and voxel-based segmentation results are converted into each other.

Nonetheless, it remains a critical challenge to preserve the integrity and continuity of the coronary artery tree due to the existence of the fragmentation vessels. Additionally, the existing segmentation methods of integrating mesh deformation networks are limited to big organs with regular shapes, such as the liver and hippocampus. It is hard to generate complete and elaborate meshes of the coronary artery from voxel-based segmented results, due to its intricated structures and narrow branches.

To cope with the above problems, we propose a novel geometry-based segmentation network, where the generated vectorized mesh of the coronary artery becomes more integrated compared to the voxel-based segmentation results. Furthermore, the mesh results of the coronary artery are smoother with plentiful details, particularly in tiny and narrow branches. Extensive experiments confirm the robustness and feasibility of our method.

## 2  METHODOLOGY

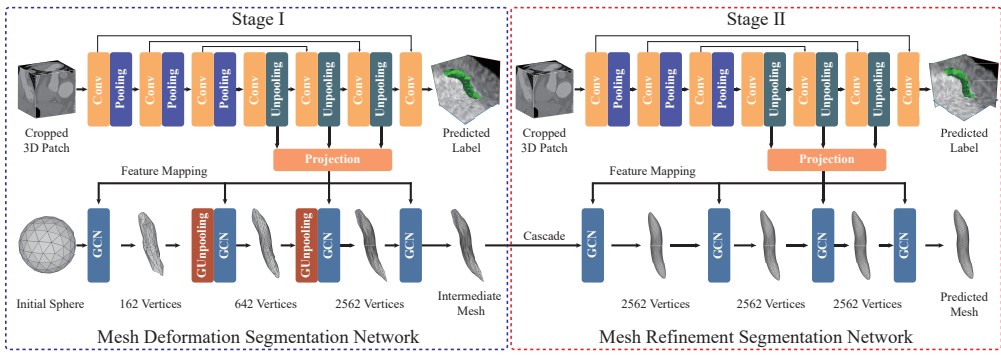

Figure 1: Our geometry-based end-to-end cascade segmentation network for generating mesh of the coronary artery.

Aiming at constructing the mesh of the coronary artery directly, an end-to-end cascade neural network is designed and illustrated as shown in Fig.1. At stage I, a classical U-shaped neural network extracts image features in CCTA, which are projected into the mesh space. Guided by these projected features, the initial sphere mesh is deformed toward the ground truth through the graph convolutional neural network (GCN). The mesh will add more vertices by graph unpooling to improve the details during the deformation. Besides, the segmentation neural network outputs voxel-based results and vectorized mesh results, simultaneously. The whole neural network is driven by voxel-based results and mesh results of the coronary artery. At stage II, refinement is performed on the vectorized mesh predicted by stage I. Without the need for graph unpooling, the GCN deforms the mesh to supplement more details of the coronary artery. The parameters of the U-shaped neural network are fixed and copied from the first stage, guaranteeing that the extracted image features are the same in both stages. The total loss consists of various losses to drive the U-shaped network and GCN as shown in Eq.1.

$$\mathcal{L} = \lambda_1 \mathcal{L}_{Dice} + \lambda_2 \mathcal{L}_{CE} + \lambda_3 \mathcal{L}_{CD} + \lambda_4 \mathcal{L}_{Lap} + \lambda_5 \mathcal{L}_{NC} + \lambda_6 \mathcal{L}_{EG} \tag{1}$$

where $\lambda_{1-6}$ represents the weight of each loss. Dice and Cross-Entropy (CE) compute the loss of voxel-based results. Meanwhile, Chamfer Distance (CD) calculates the loss of point clouds between prediction and ground truth, and laplacian smoothing (Lap), normal consistency loss (NC) and edge loss (EG) are used to regularize it.

However, the intricate structure of the coronary artery exhibits a great challenge for the neural network. GCN is hard to learn such sophisticated and elaborate morphology. Accordingly, the cropped mesh of the coronary artery is classified into two categories: tube and bifurcation. Compared with twisted, irregular and multi-forks coronary artery mesh, tube and bifurcation have simpler morphology, which is more straightforward to be learned by the neural network. Hence, morphological reg-

ularization is devised to regularize cropped mesh into the structure of tube or bifurcation. Through regularized meshes, the geometry-based neural networks could learn the structural and morphological features of the coronary artery more easily and precisely.

On the other hand, we generate accurate ground truth meshes of the coronary artery by our centerline-based approach, instead of the traditional marching cube method. Along the centerline, obtained by skeletonizing the mask of the coronary artery, every branch of the mesh is reconstructed with a smooth and delicate surface. Given the mesh of each branch, the mesh boolean union operation is implemented to merge them and finish the complete mesh of the coronary artery. Our approach achieves the performance of reconstructing the ground truth meshes of the coronary artery, with integral structure and abundant details of tiny and narrow branches. In consequence, the reconstructed ground truth meshes of the coronary artery bring considerable improvement to the geometry-based segmentation neural network.

## 3 EXPERIMENTS

In this section, the dataset and evaluation metrics are first introduced. Then, extensive experiments are conducted, evidencing the viability and practicality of coronary artery segmentation results generated by our model.

| Methods | Dice | HD | Smoothness | Num of Segments | Chamfer Distance |
|---|---|---|---|---|---|
| ResUnet | 0.575 | 3.960 | 0.550 | 116.4 | 111.85 |
| H-DenseUnet | 0.587 | 5.662 | 0.537 | 113.1 | 195.47 |
| Unet3D | 0.633 | 1.886 | 0.585 | 60.9 | 64.05 |
| nnUNet | 0.743 | 0.779 | 0.791 | 15.8 | 34.90 |
| FFNet | 0.707 | 6.026 | 0.729 | 126.6 | 29.80 |
| 3D-FFR-Unet | 0.770 | 0.785 | 0.795 | 129.6 | 6.59 |
| Voxel2Mesh | 0.191 | 28.861 | 0.062 | 2.0 | 519.61 |
| **Ours** | **0.779** | **0.299** | **0.050** | **2.0** | **2.82** |

Table 1: Quantitative Evaluation Results of the Coronary Artery Segmentation for Different Methods on CCA-520 Dataset.

### 3.1 DATASET AND EVALUATION

Our proposed method is verified on our collected dataset CCA-520, which consists of 520 cases with coronary artery disease. To validate our model in small-scale data, comparative experiments are designed: 20 cases are used for training, and 500 cases for testing. Ground truth masks of 520 cases are coronary artery internal diameter annotations labelled by four radiologists.

Various metrics are applied to assess the performance of different models. Dice evaluates the intersection of segmentation results and ground truth. Hausdorff distance (HD) and chamfer distance measure the morphological difference. Smoothness judges the flatness of the segmented reconstruction mesh by computing the normal consistency for each pair of neighboring faces of the reconstruction mesh. Furthermore, for assessing the integrity and continuity of the coronary artery, the metric Num of Segments is proposed to count the number of connecting vessels.

Several methods are selected for comparison experiments, namely ResUnet Zhang et al. (2018), H-DenseUnet Li et al. (2018), Unet3D Çiçek et al. (2016), nnUNet Isensee et al. (2021), FFNet Zhu et al. (2022), 3D-FFR-Unet Song et al. (2022), Voxel2Mesh Wickramasinghe et al. (2020). They belong to three main types of coronary artery segmentation, which are 2D pixel-based, 3D voxel-based and geometry-based segmentation methods, respectively.

### 3.2 RESULTS AND DISCUSSION

Table.1 displays the quantitative evaluation results of the coronary artery segmentation for different methods on CCA-520 dataset. In terms of the similarity to ground truth, our method achieves Dice of 0.779 and hausdorff distance of 0.299, exceeding other segmentation methods of different types. The

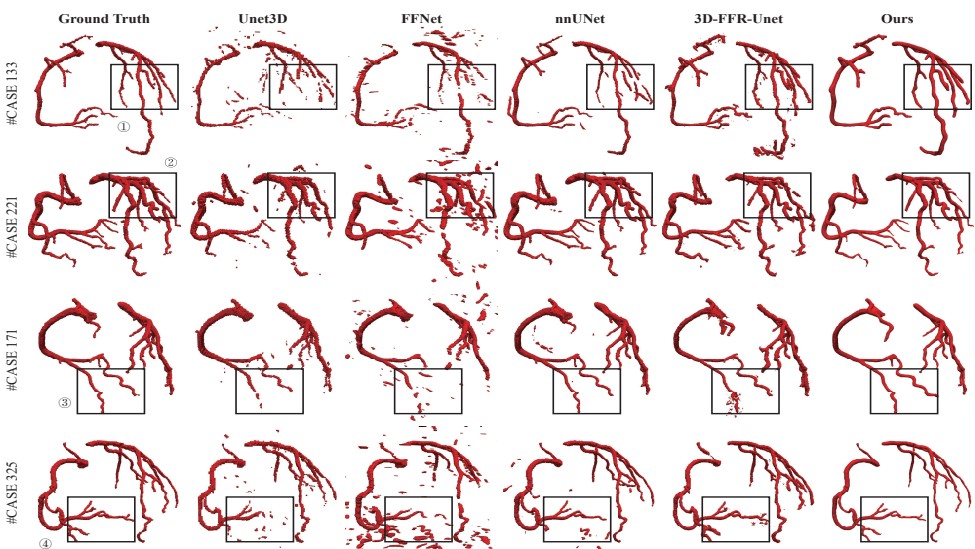

Figure 2: Coronary Artery Segmentation Results on CCA-520 Dataset.① and ②: Complex multi-forks of the coronary artery. ③ and ④: Tiny and narrow branches

high Num of Segments of voxel/pixel-based segmentation methods indicates that they are severely interfered with by fragmentations of vessels, missing intact and complete structure of the coronary artery. The limitation of the sparsity also brings the disadvantage of low smoothness. Voxel2mesh predicts the organ as a whole, and the initial spheres are difficult to deform into complex branches of the coronary artery, resulting in a particularly low Dice. Faced with the complicated structure of the coronary artery, our approach can cope well with the segmentation task, generating accurate labels of the coronary artery. Besides, the geometry-based segmentation network outputs vectorized mesh of the coronary artery directly, bringing the highest smoothness in the reconstruction mesh of the coronary artery. Moreover, the geometry-based segmentation forms the mesh of the coronary artery by deforming initial spheres, so that no fragmentations of segmented vessels occur as with the voxel-based methods. The entire coronary artery tree is produced completely and elaborately. As for chamfer distance, it reveals our generated mesh of the coronary artery has similar morphology to the ground truth of the coronary artery with intricated morphology.

Fig.2 depicts the coronary artery segmentation results of different methods on our collected CCA-520 dataset. Voxel-based segmentation results mostly are hampered by fragmentations of vessels, missing an intact coronary artery structure. Conversely, our model produces meshes of the coronary artery with a complete structure, smooth multi-forks and clear tiny branches. The geometry-based segmentation network of the coronary artery elegantly avoids the fragmentations of segmented vessels and generates intact and continuous meshes of the coronary artery. Due to the vectorized characteristic of the mesh, the tiny and narrow branch end of the coronary artery can be more accurately delineated, eliminating the limitations of sparsity and the low resolution of CCTA images. By simplifying the training of our geometry-based segmentation network through morphological regularization, our model generates natural and smooth transitions at multi-forks of the coronary artery. In summary, the generated mesh of the coronary artery achieves superior performance in terms of accuracy, smoothness and integrity.

## 4 CONCLUSION

In this paper, we propose a geometry-based end-to-end segmentation model for the coronary artery tree with a complicated structure. The segmentation network is capable of generating precise, intact and smooth meshes, absent fragmentations of segmented vessels. Extensive experiments demonstrate our model, with a Dice of 0.779 on our CCA-520 dataset, surpassing other mainstream methods.

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
