# OpenReview forum: "GEOMETRY-BASED END-TO-END SEGMENTATION OF CORONARY ARTERY IN COMPUTED TOMOGRAPHY ANGIOGRAPH"
_ICLR.cc/2023/Workshop/TML4H — ICLR 2023 Workshop TML4H Poster_

### Official Review · Reviewer_V59A · 2023-02-23
**Reasonable framework and good motivation**

**Rating:** 6
**Confidence:** 5

**Review:**

This paper proposed an end-to-end cascaded model for artery segmentation from CTA data. The authors embedded a mesh generation part to help the model training, significantly improving the topology-related performance. The overall logic flow is clear and straightforward. The proposed model also outperforms other models. There are still some concerns, which may be helpful to improve the paper further.

1. The cascaded design is not new in the vessel segmentation task. Some related works should be discussed as a supplement.
2. This paper explicitly added a shape-related regularzation. The ablation study is required to demonstrate the ability of each design. How is the performance of the proposed model without the mesh generation part?
3. The authors are suggested to make the code public.
4. What is the detail of the GCN part?
5. Does this paper match the scope of the TML4H workshop? It seems that this paper is a typical paper for fully supervised medical image processing.

Overall, considering the excellent motivation and solution, this paper is recommended as "6: Marginally above acceptance threshold".

---

### Official Review · Reviewer_vBZC · 2023-02-28

**Rating:** 6
**Confidence:** 4

**Review:**

Authors present a  geometry-based end-to-end segmentation method for the coronary artery in computed tomography angiography. The method has been described without any equations to support the formulations. The description hints at a interesting method but equations help to make it clear.
Experiments are fine and the results are good.

---

### Meta-Review · Area_Chair_FJ89 · 2023-03-03

**Recommendation:** Accept (Poster)
**Confidence:** 4

**Metareview:**

This paper presented a coronary artery segmentation algorithm from computed tomography angiography. The algorithm combines the voxel-based segmentation and geometry-based segmentation to enforce the topology structure. The motivation is very clear and the results look promising. Meanwhile, the paper is missing many tech details which increases the difficulty to reproduce and verify its effectiveness. It will be great if the author could add more equations and algorithm details in the supplementary of the final version.